# Profoundly Disturbed Lipoproteins in Cirrhotic Patients: Role of Lipoprotein-Z, a Hepatotoxic LDL-like Lipoprotein

**DOI:** 10.3390/jcm11051223

**Published:** 2022-02-24

**Authors:** Eline H. van den Berg, Jose L. Flores-Guerrero, Eke G. Gruppen, Erwin Garcia, Margery A. Connelly, Vincent E. de Meijer, Stephan J. L. Bakker, Hans Blokzijl, Robin P. F. Dullaart

**Affiliations:** 1Department of Gastroenterology and Hepatology, University of Groningen, University Medical Center Groningen, 9700 RB Groningen, The Netherlands; h.blokzijl@umcg.nl; 2Department of Nephrology, University of Groningen, University Medical Center Groningen, 9700 RB Groningen, The Netherlands; joseluis.dbt@gmail.com (J.L.F.-G.); e.g.gruppen@umcg.nl (E.G.G.); s.j.l.bakker@umcg.nl (S.J.L.B.); 3Laboratory Corporation of America Holdings (Labcorp), Morrisville, NC 27560, USA; garce14@labcorp.com (E.G.); connem5@labcorp.com (M.A.C.); 4Department of Hepatopancreatobiliary Surgery and Liver Transplantation, University of Groningen, University Medical Center Groningen, 9700 RB Groningen, The Netherlands; v.e.de.meijer@umcg.nl; 5Department of Endocrinology, University of Groningen, University Medical Center Groningen, 9700 RB Groningen, The Netherlands; dull.fam@12move.nl

**Keywords:** cirrhosis, lipid profile, NMR, HDL-cholesterol, LP-Z, orthotopic liver transplantation

## Abstract

Detailed information regarding lipoprotein concentrations and subfractions in cirrhotic patients before and after orthotopic liver transplantation (OLT) is lacking. Lipoprotein-Z (LP-Z) is a recently characterised abnormal, hepatotoxic free cholesterol-rich low-density lipoprotein (LDL)-like lipoprotein. We determined the lipoprotein profiles, including LP-Z, in cirrhotic patients and OLT recipients and assessed the prognostic significance of LP-Z on the OLT waiting list. We performed analyses in cirrhotic transplant candidates and non-cirrhotic OLT recipients. A population-based cohort was used as reference. The setting was a University hospital. Lipoprotein particle concentrations and subfractions were measured by nuclear magnetic resonance spectroscopy. In the cirrhotic patients (N = 130), most measures of triglyceride-rich lipoproteins (TRL), LDL, and high-density lipoproteins (HDL) were much lower compared to the OLT recipients (N = 372) and controls (N = 6027) (*p* < 0.01). In the OLT recipients, many lipoprotein variables were modestly lower, but HDL-cholesterol, triglycerides, and TRL and HDL size were greater vs. the control population. LP-Z was measurable in 40 cirrhotic patients and 3 OLT recipients (30.8% vs. 0.8%, *p* < 0.001). The cirrhotic patients with measurable LP-Z levels had profoundly lower HDL-cholesterol and particle concentrations (*p* < 0.001), and worse Child Pugh Turcotte classifications and MELD scores. The presence of LP-Z (adjusted for age, sex, and MELD score) predicted worse survival in cirrhotic patients (HR per 1 LnSD increment: 1.11, 95%CI 1.03–1.19, *p* = 0.003). In conclusion, cirrhotic patients have considerably lower plasma concentrations of all major lipoprotein classes with changes in lipoprotein subfraction distribution. After OLT, these lipoprotein abnormalities are in part reversed. LP-Z is associated with cirrhosis. Its presence may translate in disturbed HDL metabolism and worse survival.

## 1. Introduction

The liver plays a central role in lipoprotein metabolism [1,2,3]. The liver produces very low-density lipoproteins (VLDL) and high-density lipoproteins (HDL), as well as enzymes which are critically involved in extrahepatic lipoprotein metabolism, such as lecithin cholesterol acyltransferase (LCAT), an enzyme that is involved in the metabolism of HDL by its ability to esterify free cholesterol to cholesteryl esters [4,5] and hepatic lipase (HL), which is able to hydrolyze triglycerides in various lipoproteins [5]. Important mechanisms which contribute to the pathogenesis of lipoprotein abnormalities in severe chronic liver diseases include a decrease in (apo)lipoprotein synthesis, impaired degradation, altered uptake of lipoprotein complexes by the liver, as well as LCAT deficiency [2,3,6,7,8]. In cirrhosis, low plasma levels of total cholesterol, as well as triglycerides, are common [9]. Previous studies have also described low levels of VLDL, low-density lipoprotein cholesterol (LDL-C), HDL cholesterol (HDL-C), and apolipoprotein B (apoB) and apolipoprotein A-I (apoA-I), which are major apolipoproteins associated with VLDL, LDL, and HDL, respectively [3]. In advanced stages of liver disease, circulating levels of all major lipoprotein classes decline [10,11,12], but variable patterns of lipoprotein changes exist, probably related to liver disease aetiology and concomitant comorbidities [3,12].

In patients with end-stage liver disease, orthotopic liver transplantation (OLT) is the only available curative treatment [13]. After OLT, recipients often develop dyslipidaemia with higher total and LDL-C concentrations [14,15,16], which may lead to an increased occurrence of cardiovascular and metabolic comorbidities, including atherosclerotic manifestations, heart failure, and type 2 diabetes (T2D) [14,17,18,19]. Post-OLT, hypercholesterolemia has been reported in 31% at 1 year, 46% at 2 years, and 37% after 3 years [14,20,21,22]. Corticosteroids are major determinants of increased cholesterol levels, and early withdrawal of prednisolone decreases the incidence and severity of hypercholesterolemia after OLT [15,23,24]. Additionally, dyslipidaemia post-transplantation is exacerbated by mammalian target of rapamycin (mTOR) inhibitors [25].

Despite many studies on lipoprotein abnormalities in severe chronic liver disease, no data have been currently reported concerning nuclear magnetic resonance (NMR)-determined alterations in lipoprotein subfractions in patients with liver cirrhosis or in OLT recipients. Availability of such data is clinically relevant in view of differential associations of LDL, VLDL, and HDL subfractions and particle concentrations with incident atherosclerotic cardiovascular disease and T2D [26,27,28,29,30,31,32]. Of particular relevance, an abnormal LDL-like lipoprotein, designated Lipoprotein-Z (LP-Z), has been recently characterised using NMR methodology [33]. LP-Z is probably similar to an abnormal lipoprotein sometimes present in patients with obstructive jaundice [33,34]. Similar to LDL, LP-Z contains apoB but carries more free cholesterol, phosphatidyl choline, and triglycerides, and has a reduced cholesteryl ester content compared to LDL [33]. Given that the characterisation of LP-Z by NMR methodology has only been recently described, neither its pathogenesis nor its metabolic and clinical corollaries are yet imminent. Presence of LP-Z in plasma has been found in some patients with obstructive jaundice, hypertriglyceridemia, and alcoholic liver disease [9,33,34]. Interestingly, LP-Z could so far not be detected in healthy subjects, nor in subjects with primary biliary cholangitis (PBC), primary sclerosing cholangitis (PSC), autoimmune hepatitis (AIH), or T2D [33].

We initiated the present study firstly to characterise distinct NMR lipoprotein profiles in cirrhotic patients with end-stage liver disease screened for OLT and in OLT recipients (TransplantLines cohort study), and in a general population-based control cohort. Secondly, we determined to which extent LP-Z is detectable in plasma from cirrhotic patients compared to stable OLT recipients and to characterise the degree of lipoprotein abnormalities according to LP-Z presence in plasma. Thirdly, we interrogated whether detectable LP-Z may have prognostic significance, in addition to the Model for End-Stage Liver Disease (MELD) score in cirrhotic patients.

## 2. Materials and Methods

### 2.1. Study Population

This study was performed among participants of the TransplantLines cohort study, a large prospective transplant biobank and cohort study carried out in the University Medical Center Groningen, the Netherlands (NCT03272841) [35]. The TransplantLines study was approved by the Medical Ethics Committee of the University Medical Center Groningen (METc 2014/077) and is performed in accordance with the guidelines of the Declaration of Helsinki. All participants provided written informed consent. TransplantLines is a prospective biobank and cohort study, which aims to provide a better understanding of the causes of disease-related and ageing-related outcomes and health problems, both physical and psychological, in patients with end-stage disease screened for solid organ transplantation and solid organ transplant recipients. All transplant recipients, including heart, lungs, kidney, liver, and small bowel, were invited to participate in the TransplantLines study. The exclusion criteria were no mastery of the Dutch language and inability to intellectually understand questionnaires or physical tests [35].

This report follows the Strengthening the Reporting of Observational Studies in Epidemiology (STROBE) reporting guideline. The present study was a cross-sectional analysis of patients in the liver transplant subgroup of TransplantLines. For the present study, data from: 1) pre-transplant cirrhotic patients and 2) OLT recipients at least 3 months after OLT, with available NMR lipoprotein measurements, were used. This resulted in 502 participants (flow sheet of participants: see Figure 1). Of the participants with cirrhotic end-stage liver disease (N = 130), 15 patients were prospectively studied over a period ranging from 3 to >24 months post liver transplantation. Of these, 14 patients were also included in the non-cirrhotic OLT recipient group (N = 372), because 1 patient developed cirrhosis post-OLT.

As a control group, the study population of the Prevention of REnal and Vascular ENd-stage Disease (PREVEND) cohort study was used [36,37]. The PREVEND study was approved by the Medical Ethics Committee of the University Medical Center Groningen (MEC96/01/022) and is performed in accordance with the guidelines of the Declaration of Helsinki. Written informed consent was obtained from all participants. PREVEND is a large prospective general population-based study, which was initiated to investigate cardiovascular and renal disease. All inhabitants (28 to 75 years old) of Groningen, the Netherlands were sent a questionnaire on demographics and cardiovascular morbidity. Pregnant women, type 1 diabetic subjects, and T2D subjects using insulin were excluded from participation [36,37]. For the present study, we used the data of participants as a control group who completed the second screening round in the PREVEND study, in which NMR lipoprotein data were available (N = 6027).

### 2.2. Data Collection and Measurements

TransplantLines has a continuous data collection which started on June 2015. For the present study, data were collected up to December 2020. During visits at the outpatient clinic, questionnaires and blood samples were collected from all participants according to the TransplantLines protocol. At the time of blood collection, patients continued their regular medication. A standardized protocol was used to obtain anthropometric measurements. Patient information, including weight, height, body mass index (BMI), smoking status, medication use (glucose and lipid lowering drugs), and medical history, such as cardiovascular disease (CVD) and diabetes, was extracted from the TransplantLines Biobank. Additional review of all electronic patient records of study participants was performed in order to retrieve relevant additional data concerning the aetiology of liver disease, non-cirrhotic or cirrhotic state (based on imaging, histology, or transient elastography) at the time of NMR lipoprotein measurements, and biochemical and clinical variables to calculate the MELD and Child Pugh Turcotte (CPT) scores. To assess the severity of chronic liver disease, the CPT and MELD scores were used. The MELD score was calculated by serum bilirubin, creatinine, and international normalized ratio (INR) [38,39]. The CPT score was calculated by total bilirubin, serum albumin, INR, ascites, and hepatic encephalopathy [40].

In the PREVEND cohort, baseline data were collected on demographics, lifestyle factors, anthropometric measurements, medical history, and medication use, which was combined with information from a pharmacy-dispensing registry as previously described [36,37].

### 2.3. Laboratory Analysis

In TransplantLines, all blood samples were collected by experienced nurses at the outpatient clinic after an overnight fast. Alanine aminotransferase (ALT), aspartate aminotransferase (AST), gamma-glutamyl transferase (GGT), alkaline phosphatase (ALP), total and direct bilirubin, albumin, glycated haemoglobin (HbA1c), and fasting glucose were analysed with standardized laboratory measurements and quality assessment control at the Department of Laboratory Medicine of the University Medical Center Groningen [35]. In PREVEND, venous blood samples were drawn after an overnight fast while participants rested for 15 minutes. Laboratory methods for PREVEND are reported as described in detail previously [36,37,41].

Ethylenediaminetetraacetic acid (EDTA) anticoagulated plasma and serum samples were obtained by centrifugation at 1400× *g* for 15 min at 4 °C. EDTA-anticoagulated plasma samples for lipid and lipoprotein profiles in TransplantLines and PREVEND were stored at −80 °C until analysis and were measured using a Vantera Clinical Analyzer (Labcorp, Morrisville, NC, USA), a fully automated, high-throughput, 400 MHz proton (1H) NMR spectroscopy platform. Plasma samples were prepared on board the instrument, and automatically delivered to the flow probe in the NMR spectrometer’s magnetic field [42]. Data acquisition on the Vantera and the spectra data processing have been reported in greater detail elsewhere [43,44]. Lipoprotein parameters were measured by NMR spectroscopy at Labcorp on plasma EDTA specimens [43,44,45]. Total cholesterol, HDL-C, triglycerides, apoB, and apoA-I were measured employing the LP4 algorithm [29,30,46]. The %CV for apoB ranges from 1.1 to 2.4 and for apoA-I ranges from 1.4 to 2.2. LDL-C was calculated using the Friedewald equation. Non-HDL cholesterol was calculated as total cholesterol minus HDL-C. Very large, large, medium, small, and very small triglyceride rich lipoprotein particles (TRLP) and large, medium, and small LDL particles (LDLP) were quantified using the conventional deconvolution method and the amplitudes of their spectroscopically distinct lipid methyl group NMR signals [29,31,44,45]. Total TRLP were calculated as the sum of the concentrations of very large, large, medium, small, and very small TRLP. Total LDLP were calculated as the sum of all LDL subfractions. Mean TRL and LDL size were calculated using the weighted averages derived from the sum of the diameters of each subfraction. Estimated ranges of particle diameter for the TRL and LDL subfractions were as follows: very large TRLP, 90–240 nm; large TRLP, 50–89 nm; medium TRLP, 37–49 nm; small TRLP, 30–36 nm; very small TRLP, 24–29 nm; large LDLP, 21.5–23 nm; medium LDLP, 20.5–21.4 nm; and small LDLP, 19–20.4 nm. The %CV for TRLP ranges from 4.5 to 11.7 and for LDLP ranges from 1.3 to 3.2. Total HDL particles (HDLP) were calculated by the sum of the concentrations of small, medium, and large HDL particles. Mean HDL size was calculated using the weighted averages derived from the sum of the diameters of small, medium, and large HDLP multiplied by its relative mass percentage. Estimated ranges of particle diameter for the subclasses were as follows: small HDL, 7.4–8.0 nm; medium HDL, 8.1–9.5 nm; and large HDL, 9.6–13 nm. Mean diameter for the HDL subfractions were as follows: H1P, 7.4 nm; H2P, 7.8 nm; H3P, 8.7 nm; H4P, 9.5 nm; H5P, 10.3 nm; H6P, 10.8 nm; and H7P, 12.0 nm. Small HDL comprises H1 and H2, medium HDL comprises H3 and H4, and large HDL comprises H5 to H7. The %CV for HDLP ranges from 0.8 to 1.5.

LP-Z particles were quantified using the LP4 algorithm as previously described by Bedi et al. [33]. In short, the NMR signals from LP-Z span the NMR chemical shift range corresponding to small LDL (18.5–21.5 nm diameter). However, the signals for LP-Z and small LDL differ in their signal line shapes, which allows the software algorithm to distinguish LP-Z from LDL. The assay uses the LP4 deconvolution model that fits the components of the normal lipoproteins first, followed by a model that includes components for LP-Z and LP-X [33]. The %CV for LP-Z particles ranges from 1.5 to 6.7. To ensure the stability of LP-Z particles, fresh serum samples, without detectable LP-X and a large concentration of LP-Z particles, were tested via NMR and then placed at <−20 °C. The samples were thawed and retested after 1, 2, 3, and 4 freeze-thaw cycles. Unlike LP-X particles that have been shown to be unstable after a single freeze-thaw cycle, the LP-Z particle concentrations remained the same for up to 4 freeze-thaw cycles [47].

### 2.4. Statistical Analysis

Statistical analyses were performed with IBM SPSS software (version 23.0 Armonk, NY, USA: IBM Corp) and R language for statistical computing software (version 4.0.3 (2020), Vienna, Austria). Continuous variables are expressed as median with interquartile range (IQR) or as numbers with percentages. Normality of distribution was assessed and checked for skewness. Differences in variables between the pre-transplant cirrhotic patients and the OLT recipients were determined by Mann–Whitney U test, Kruskal Wallis test or by Chi-square tests where appropriate. Univariate correlations for continuous response variables were analysed by Spearman rank correlation coefficients. The difference in lipid profile, adjusted for age and sex, of the pre-transplant cirrhotic patients and OLT recipients compared to the control subjects from the PREVEND cohort study was estimated with linear regression analyses and expressed in coefficients of standard deviation (SD) scores (Z-scores). Multivariable logistic regression analyses were carried out to disclose the independent determinants of the presence of LP-Z when taking account of clinical covariates. Kaplan–Meier curves with log-rank test were performed to determine the effect of the presence of LP-Z on mortality on the waiting list, with the date of OLT as the census date. Cox proportional hazards models were used to compute hazard ratios (HRs) and 95%CI of LP-Z (expressed per 1 Ln SD increment) on mortality risk on the waiting list. For calculation purposes, zero values of LP-Z were set at 0.001 in the Cox regression analysis. HRs were calculated in models adjusted for age, sex, and MELD score. We estimated changes in risk of mortality on the waiting list by inclusion of detectable LP-Z using the MELD score as the base model with Harrell’s C-statistics [48]. To account for the number of independent comparisons, we applied a Bonferroni correction. Therefore, two-sided *p*-values of <0.01 (0.05/3) were considered statistically significant, given the use of three independent comparisons (pre-transplant cirrhotic patients, OLT recipients, and PREVEND population).

## 3. Results

### 3.1. Comparison of Clinical and Laboratory Characteristics in Pre-Transplant Cirrhotic Patients and OLT Recipients

In 130 pre-transplant cirrhotic patients and 372 OLT recipients, MAFLD and cholestatic liver disease were the most frequent primary liver diseases, accounting for >50% of indications for transplantation. Baseline characteristics, such as age, sex, smoking, history of CVD or diabetes, and drug-use, did not differ significantly between the pre-transplant cirrhotic patients and OLT recipients (Table 1). In the pre-transplant cirrhotic patients, the median MELD score was 15 (IQR 10−19), with a CPT A classification in 21.5%, CPT B classification in 49.2%, and CPT C classification in 29.2%. As expected, liver function tests were all higher in the pre-transplant cirrhotic patients with lower albumin levels compared to the OLT recipients. In the pre-transplant cirrhotic patients, the lipid profile was profoundly different compared to the OLT recipients with lower plasma total cholesterol, HDL-C, and LDL-C, non-HDL cholesterol, and lower triglycerides, apoA-I, total TRLP, and total HDLP levels (Table 1, all *p* < 0.001). Only apoB, very large and very small TRLP, LDLP, and large HDLP were not significantly different between the pre-transplant cirrhotic patients and OLT recipients after adjustment for age, sex, and primary liver disease aetiology. Moreover, TRL size was smaller, whereas LDL and HDL size were significantly greater in the pre-transplant cirrhotic patients (*p* < 0.001). The differences in TRLP and decrease in HDLP levels were more profound in the patients with higher CPT classification (Appendix A). Moreover, TRLP and HDLP levels were inversely correlated with MELD scores, whereas LDLP was positively correlated with MELD scores (Appendix A). Notably, LP-Z was present in 40 (30.8%) pre-transplant cirrhotic patients compared to only 3 (0.8%) OLT recipients (*p* < 0.001).

### 3.2. Comparison of Pre-Transplant Cirrhotic Patients and OLT Recipients with PREVEND Population

Baseline characteristics, liver function, and lipid profile for the control cohort PREVEND are presented in Appendix A. In the pre-transplant cirrhotic group, all (apo)lipoprotein measures were much lower compared to the PREVEND population, except for large LDLP, large HDLP, and HP7 (Table 2). Also, in the group of patients with MAFLD as underlying aetiology, total cholesterol, triglycerides and HDL-C as well as TRLP and HDLP were lower compared with the values in the PREVEND population (data not shown). In the OLT recipients, apoA-I, very large, large, and very small TRLP, large and small LDLP, and medium HDLP were not different compared to the PREVEND population, whereas HDL-C, triglycerides, medium TRLP, large HDLP, HP4, HP5, HP6, and HP7 were higher. The other lipoprotein measures were still lower. LDL size and HDL size were greater in the pre-transplant cirrhotic patients, whereas TRL size and HDL size were greater in the OLT recipients compared to the PREVEND population (Table 2). Notably in the PREVEND cohort, measurable LP-Z levels were only found in five participants (0.08%), with a strikingly higher regression coefficient of the *Z*-score difference of 3.176 (95% CI 3.017–3.336) in the pre-transplant cirrhotic patients (Table 2).

### 3.3. Determinants of the Presence of LP-Z

No significant differences were found with respect to the aetiology of primary liver disease in the pre-transplant cirrhotic patients, except for cholestatic liver diseases (20% vs. 40%, *p* < 0.05) between those with and without measurable LP-Z (Table 3). Cirrhotic patients with measurable LP-Z levels did have higher CPT classifications and MELD scores with concomitantly higher ALT, ALP, and total and direct bilirubin levels (*p* < 0.05). In the cirrhotic patients, LP-Z was more frequently present with higher CPT classification (Figure 2A) and was correlated with higher MELD scores (Figure 2B). In multivariable binary logistic regression analyses, MELD scores were independently associated with the presence of LP-Z in the cirrhotic patients, even when adjusted for CPT classification and the use of glucose or lipid lowering drugs (Appendix A). Furthermore, in subsequent multivariable analysis, the independent positive association of higher MELD scores with measurable LP-Z levels remained present after adjustment for aetiology of liver cirrhosis (Appendix A). In this analysis, the presence of LP-Z was positively associated with cholestatic liver disease as aetiology of cirrhosis.

In the pre-transplant cirrhotic patients, strikingly lower levels of HDL-C (0.2 vs. 1.1 mmol/L, *p* < 0.001), apo A-I (20.5 vs. 68.5 mg/dL, *p* < 0.001), and smaller HDL size (9.5 vs. 10.8 nm, *p* < 0.001) were found in the presence of LP-Z (Table 3). The pre-transplant cirrhotic patients with detectable LP-Z also had lower TRLP levels, higher LDLP levels, and strongly reduced HDLP levels (Figure 3). When adjusted for age, sex, and MELD score, TRLP was not independently associated with the presence of LP-Z (Appendix A; TRLP Model 3). The presence of LP-Z was however independently associated with higher levels of LDLP (Appendix A; LDLP Model 3, *p* < 0.001) and lower levels of HDLP (Appendix A; HDLP Model 3, *p* = 0.001). Finally, follow-up data were available in 15 patients. In 14 out of 15 cirrhotic patients, LP-Z was measurable pre-transplantation and only 1 patient had a measurable LP-Z value post-transplantation.

### 3.4. Presence of LP-Z and Mortality in Pre-Transplant Cirrhotic Patients

Twenty-nine of the pre-transplantation cirrhotic patients died while on the waiting list (Table 1). Out of 40 patients with detectable LP-Z, 11 (27.5%) patients died vs. 18 (20.0%) patients without detectable LP-Z (Table 3). The presence of LP-Z, adjusted for age, sex, and MELD score, predicted a worse survival in pre-transplant cirrhotic patients (Figure 4: Kaplan–Meier plot; log-rank test: *p* < 0.001). In Cox proportional hazard regression analyses that examined LP-Z concentrations as HR per 1 Ln SD increment, increased LP-Z concentrations were associated with increased risk of mortality on the waiting list in the crude analyses (HR 1.14, 95%CI 1.07–1.21, *p* < 0.001). The association remained present in analyses adjusted for age, sex (HR 1.15, 95%CI 1.08–1.23, *p* < 0.001), and MELD score (HR 1.11, 95%CI 1.03–1.19, *p* = 0.003). A mortality risk prediction model containing MELD score, as well as age and sex, yielded a C-index of 0.80 (95%CI 0.74–0.85). Notably, the addition of detectable LP-Z improved the C-index to 0.82 (95%CI 0.77–0.86, *p* = 0.01).

## 4. Discussion

Using novel NMR methodology, we have demonstrated considerably lower levels of all major lipoprotein classes, including lower plasma concentrations of TRLP, LDLP, and HDLP, in the pre-transplant cirrhotic patients selected for transplantation screening compared to the values in the large population-based PREVEND cohort. The HDL particle concentration and apoA-I levels were most profoundly diminished. In the OLT recipients, most lipoprotein variables approximately returned to the values found in the PREVEND cohort. However, plasma total cholesterol, LDL-C, non-HDL cholesterol, apoB, as well as TRLP, LDLP, and HDLP particle concentrations were still lower, whereas TRL size and HDL size were greater in the OLT recipients than in the PREVEND cohort. These lipoprotein abnormalities may have relevance for the occurrence of morbidities after transplantation, including atherosclerotic cardiovascular disease, heart failure, and diabetes. Notably, our study demonstrated for the first time that LP-Z, an abnormal LDL-like lipoprotein, was remarkably frequently detectable in pre-transplant cirrhotic patients with a prevalence of 30.8%. In contrast, in the general population, plasma LP-Z was detectable in only a very few participants. Remarkably, plasma LP-Z was detectable in the pre-transplant cirrhotic patients with a variety of underlying aetiologies, although detectable LP-Z tended to be more prevalent in the cirrhotic patients with underlying cholestatic liver diseases. Profoundly lower levels of HDL-C, apoA-I, and HDLP concentrations were found in the presence of LP-Z. Moreover, the presence of LP-Z in plasma was associated with a worse CPT classification and a higher MELD score. Given that the median MELD score among patients selected for OLT in the current study was not very high (15 points) it can be surmised that lipoprotein abnormalities, in particular low HDL-C and low HDLP could be even more pronounced in cirrhotic patients with more advanced disease. In the OLT recipients, LP-Z was detectable much less frequently, and plasma LP-Z disappeared in most patients during follow-up. Taken together, the present study reveals for the first time the presence of plasma LP-Z in a cohort of pre-transplant cirrhotic patients in association with cirrhosis severity. Moreover, our current findings conceivably suggest that a detectable LP-Z may translate in a disturbed HDL metabolism. Strikingly in the pre-transplant cirrhotic patients, the presence of LP-Z was associated with worse survival on the waiting list.

The liver plays a crucial role in the synthesis, secretion, degradation, and storage of lipids and lipoproteins. Plasma lipid and lipoprotein levels generally tend to decrease along with the severity of liver disease [1,49]. Lipoprotein profiles in liver disease are however not uniform and this disparity could be explained by the different aetiologies of chronic liver diseases. In obstructive jaundice hyperlipidaemia, which is predominantly due to elevated LDL and triglycerides, is a constant finding and this appears to be the result of reabsorption of phospholipids from an obstructed biliary tree [1,50]. In cholestatic liver diseases, there is also an abnormal receptor-mediated uptake of lipoproteins from plasma [49]. Cholesterol taken up by hepatocytes suppresses hydroxymethylglutaryl-CoA-reductase, activates acyl CoA (cholesteryl acyl transferase), and inhibits expression of LDL receptors, but leaves apolipoprotein E receptor activity unchanged [49]. ApoA-I and apoA-II, major apolipoproteins that are carried by HDL, have been described to be decreased in cholestatic liver diseases together with impaired HL, lipoprotein lipase (LPL), and LCAT activities [9,49]. Hepatocytes are also able to produce protein factors involved in antioxidant defense and lipid transfer proteins, such as phospholipid transfer protein (PLTP), which affects triglyceride assembly and secretion and converts HDL in larger to smaller HDLP [51,52,53,54,55], and cholesteryl ester transfer protein (CETP) [52], which transfers cholesteryl esters from HDL towards lipoproteins of lower density. Plasma lipoproteins also tend to decrease in parenchymal liver diseases, where abnormalities in the composition of various lipoproteins depends on disturbed apolipoprotein synthesis, lipoprotein receptor abundance, and expression of enzymes involved in lipoprotein metabolism [1].

The present study was carried out in a considerable number of 130 cirrhotic patients. In comparison, previous studies in cirrhosis reported detailed lipoprotein measurements in up to 153 patients [3,56] and one research letter shortly described the basic lipid profile of 186 cirrhotic patients [12]. In these studies, NMR techniques for assessment of an elaborate lipid profile were not used. To the best of our knowledge, this study is the first describing the detailed NMR lipoprotein profiles in end-stage cirrhotic patients. In general, our current data corroborate with these earlier studies, which reported that patients with pre-transplant cirrhosis have lower levels of HDL-C, apoA-I, and total HDLP levels, as well as greater HDL size. The decrease in HDL-C and HDLP concentration despite a decrease in triglycerides and TRLP concentration, as well as the greater HDL size despite a decrease in HDL-C, clearly underscore the profound and complex alterations in lipoprotein metabolism in end-stage liver disease [31,57]. Additionally, the strong inverse relationships of the HDLP concentration with CPT classification and MELD score indicate the impact of severe end-stage liver disease on HDL metabolism. In contrast to end-stage liver disease, OLT recipients are often hyperlipidemic [14,15,16]. Interestingly, in the currently studied OLT recipients, only HDL-C, triglycerides, medium TRLP, large HDLP, HP4, HP5, HP6, and HP7 were higher compared to the PREVEND cohort. In the OLT recipients, apoA-I, very large, large, and very small TRLP, large and small LDLP, and medium HDLP were not different and surprisingly all other lipoprotein measures were still lower compared to the PREVEND population. However, it should be noted that the post-transplant participants were only studied with a starting point of 3 months after OLT. As corticosteroid use is a major determinant of increased cholesterol levels post-transplantation [15,23,24], the difference in lipid profile in the OLT recipients could be explained by the current post-transplant immunosuppressive strategies with, for instance, the swift tapering of corticosteroids and continuation of tacrolimus treatment.

Our findings may have relevance for cardiometabolic risk in liver transplant recipients. Cholesterol loaded VLDL remnants, LDL-C, and, in particular, smaller-sized LDLP are positively associated with incident atherosclerotic cardiovascular disease, whereas HDL-C and, in particular, the HDLP concentration are inversely associated with incident atherosclerotic cardiovascular disease [27,37]. Moreover, increased small HDLP may predict higher left ventricular ejection fraction after myocardial infarction [58]. On the other hand, in patients undergoing heart catheterization, lower HDLP concentration and smaller-sized HDLP are associated with increased mortality and heart failure, an important complication after OLT [19]. In addition, evidence is accumulating that larger TRLP are positively associated with incident diabetes, whereas smaller-sized LDLP and HDLP are inversely associated with incident diabetes, even independent of fasting insulin and plasma glucose [29,31], which was not significantly different in OLT recipients compared to pre-transplant cirrhotic patients. Collectively, these findings underscore the complex and partly opposing influence of lipoprotein subfractions and sizes on major morbidities after OLT and warrant large scale longitudinal studies on this issue.

In 1976, Kostner et al. were the first to describe abnormal LDL particles that occurred in patients with obstructive jaundice and classified them into three subtypes named lipoprotein-X (LP-X), lipoprotein-Y (LP-Y), and lipoprotein-B (LP-B) [34]. Since then, only a few studies have been published that further address the characterisation of these abnormal LDL particles, which is probably due to the fact that conventional labour-intensive techniques are required to characterise these particles. Recently, LP-B was rediscovered using NMR methodology and renamed into LP-Z. A novel NMR assay was developed to further analyse the physiological and pathological signature of LP-Z [33]. In the initial paper by Bedi et al., LP-Z measured by the NMR assay could not be detected in plasma from patients with cholestatic liver diseases, such as PBC or PSC, but LP-Z was prevalent in patients with alcoholic liver disease [33]. In contrast to these findings, in the current study plasma, LP-Z was most prevalent in the cirrhotic patients with cholestatic liver diseases and was strongly related to bilirubin levels, in essence conforming to the original findings of Kostner et al. [34]. Thus, the increased presence of LP-Z in cholestatic patients could contribute to lipoprotein abnormalities in this category of cirrhotic patients. Since LP-Z was also present in other non-cholestatic aetiologies of cirrhotic liver diseases in the current cohort, the presence of LP-Z is probably not only attributable to cholestatic liver diseases, but may also be due to the presence of cirrhosis as such. In line, LP-Z was much less prevalent in the OLT recipients and our limited longitudinal observations showed that LP-Z disappeared in most patients over time.

Patients with cirrhotic end-stage liver disease have lower levels of circulating LCAT, besides alterations in CETP and HL activities [9], which raises the possibility that defective lipoprotein remodeling and catabolism may contribute to the generation of LP-Z in cirrhosis [9,49]. Very recently, it was suggested that compromised LCAT and HL activities may at least in part be responsible for the high prevalence of plasma LP-Z in alcoholic hepatitis [59]. Moreover, free cholesterol-rich reconstituted LP-Z particles were found to be hepatotoxic, as evidenced by the accelerated death of cultured hepatocytes from liver explants [59]. These findings suggest that LP-Z itself could have detrimental effects on the progression of liver disease. However, temporal cause-effect relationships are not yet clear, as is the role of LCAT in LP-Z generation and vice versa. It seems unlikely that the proposed detrimental effects of LP-Z could be explained by compromised LCAT activity alone because liver insufficiency is not a feature of genetic LCAT deficiency [60]. Additionally, the role of LP-Z as opposed to LP-X, an abnormal discoidal lipoprotein with increased free cholesterol content, which can be produced as a consequence of liver disease-induced impaired LCAT activity, has not been established. Yet, except for LDL variables (explained by the fact that LP-Z is an LDL-like lipoprotein particle and contributes to total LDL particle concentration), the presence of LP-Z was associated with much more pronounced lipoprotein abnormalities, and in particular, lower levels of HDL variables, which is likely explained at least in part by coinciding LCAT deficiency.

In the report of LP-Z in alcoholic hepatitis, LP-Z expressed as the ratio of LP-Z over VLDL + LDL apoB predicted a worse 90-day survival taking account of the MELD score [59]. In the current study, LP-Z was more closely related to the MELD score than to the CPT classification. Here, we report for the first time that in cirrhotic patients, the presence of LP-Z in plasma was independently associated with increased mortality on the waiting list. Importantly, adding LP-Z to the MELD score as the base model improved mortality risk prediction, as evidenced by a significant increase in C-statistic. Taken together, we propose that adding LP-Z may improve risk classification of cirrhotic patients, and hence may have clinical consequences regarding prioritizing patients for OLT.

The present study has several strengths. To the best of our knowledge, this is the first study to provide detailed information regarding NMR-determined particle concentrations and subfraction distribution of all major lipoprotein classes and to investigate the relationship of the presence of LP-Z in pre-transplantation cirrhotic patients and OLT recipients. Secondly, extensive data on clinical endpoints provided by the TransplantLines Biobank and Cohort study allowed us to describe our patient population in detail, assess patient survival in end-stage liver disease, and thoroughly address potential confounding factors. Limitations should also be considered. Firstly, the single-centre study design limits the external validity of our study. Secondly, robust follow-up after OLT was not available in the ongoing TransplantLines study, precluding the assessment of associations of lipoprotein characteristics with specific long-term complications. Thirdly, the presently used NMR methodology does not allow LP-X to be measured in frozen samples, as they are not stable [47].

## 5. Conclusions

Pre-transplant cirrhotic patients are featured by profoundly lower plasma TRLP, LDLP, and HDLP concentrations, as well as by abnormalities in lipoprotein subfraction distribution with smaller sized TRLP and larger sized LDLP and HDLP. After OLT, these lipoprotein abnormalities are in part reversed, with possible consequences for the development of various comorbidities. Notably, LP-Z, an abnormal LDL-like lipoprotein, is prevalent in cirrhosis, irrespective of underlying aetiology, and is strongly associated with low HDL-C, apoA-I, and HDLP concentrations. LP-Z may be a novel independent biomarker of worse survival in cirrhotic patients.

## Figures and Tables

**Figure 1 jcm-11-01223-f001:**
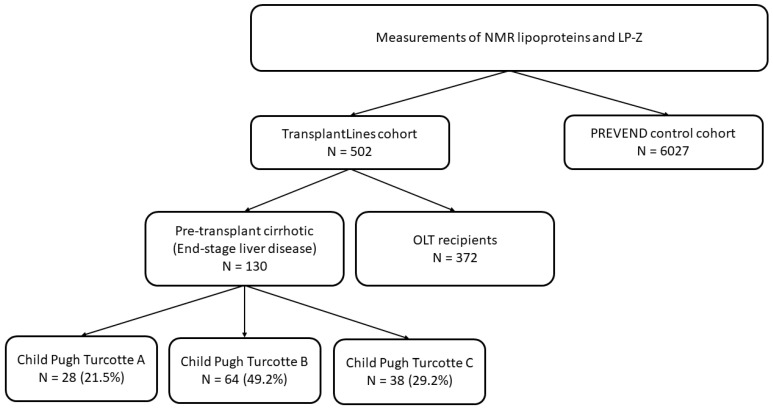
Flow chart of the study population.

**Figure 2 jcm-11-01223-f002:**
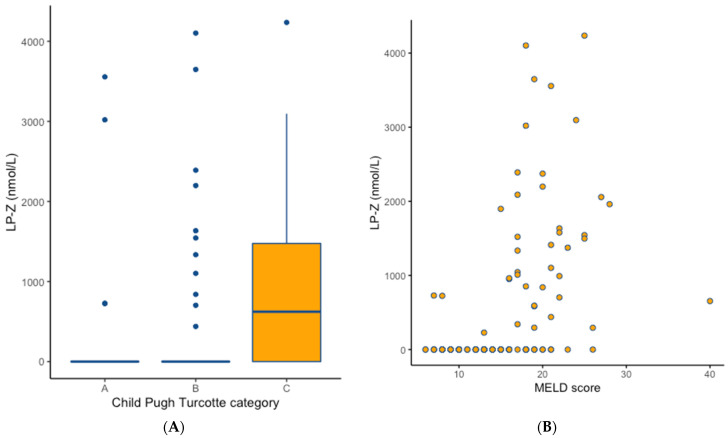
Plasma concentrations of LP-Z according to Child Pugh Turcotte classification and MELD scores in the pre-transplant cirrhotic group. (**A**): Concentrations of LP-Z according to Child Pugh Turcotte classification (Boxplots are shown and bars represent median LP-Z values). Child Pugh C classification vs. Child Pugh Turcotte Classification A and B, Kruskall–Wallis test *p* < 0.001. (**B**): Spearman rank correlation coefficient 0.591, *p* < 0.001.

**Figure 3 jcm-11-01223-f003:**
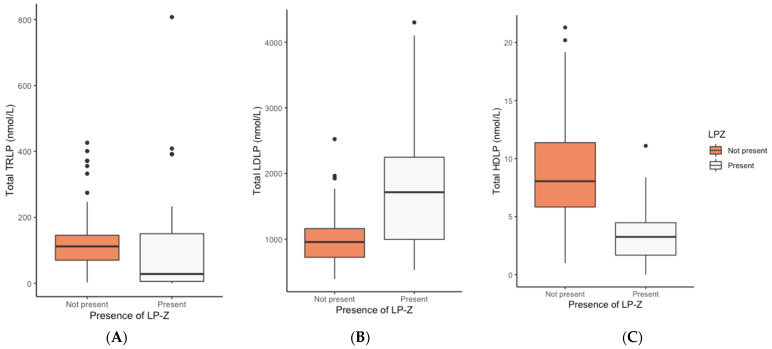
Boxplot of TRLP, LDLP, and HDLP according to presence of LP-Z in pre-transplant cirrhotic group (N = 130). Boxplot (**A**): TRLP, Boxplot (**B**): LDLP, Boxplot (**C**): HDLP. *p*-values were determined by Mann–Whitney U Test. Abbreviations: HDLP, high density lipoprotein particles; LDLP, low density lipoprotein particles; LP-Z, lipoprotein Z; TRLP, triglyceride-rich lipoprotein particles.

**Figure 4 jcm-11-01223-f004:**
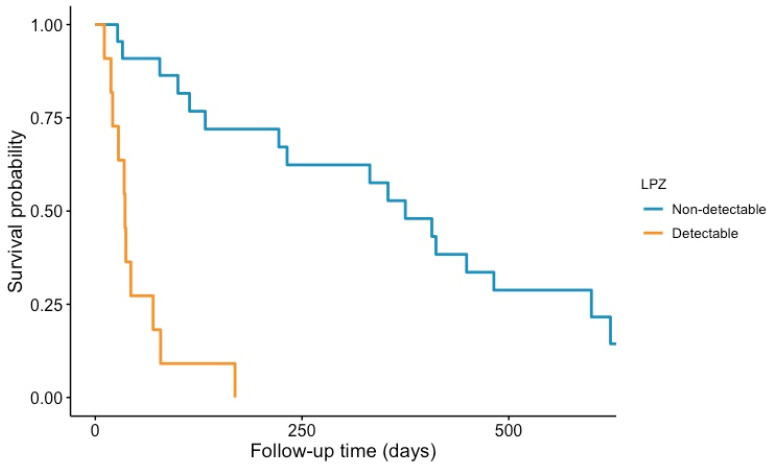
Kaplan–Meier plots for all-cause mortality in pre-transplant cirrhotic patients according to detectable plasma LP-Z concentrations. Log-rank test *p* < 0.001.

**Table 1 jcm-11-01223-t001:** Clinical characteristics and lipid profile of 130 pre-transplant cirrhotic patients and 372 orthotopic liver transplant (OLT) recipients from TransplantLines.

	Pre-Transplant Cirrhotic Patients N = 130	OLT Recipients N = 372	*p*-Value	Adjusted *p*-Value *
**Baseline characteristics**				
Sex: men/women, *n (%)*	85 (65.4)/45 (34.6)	218 (58.6)/154 (41.4)	0.174	n.a.
Age (years), *median (IQR)*	59.5 (52.0–65.0)	59.0 (47.0–67.0)	0.652	n.a.
BMI (kg/m^2^), *median (IQR)*	28.1 (24.3–30.9)	25.9 (22.9–29.6)	0.006	0.726
BMI Normal; ≤25 kg/m^2^, *n (%)* Overweight; 25–30 kg/m^2^, *n (%)* Obese; ≥30 kg/m^2^, *n (%)*	28 (30.8) 32 (35.2) 31 (34.1)	135 (41.5) 117 (36.0) 73 (22.5)	0.063 0.883 0.024	0.928 0.605 0.616
Smoking, *n (%)*	16 (18.8)	34 (11.7)	0.088	0.132
Child Pugh Turcotte classification Child Pugh Turcotte A, *n (%)* Child Pugh Turcotte B, *n (%)* Child Pugh Turcotte C, *n (%)*	28 (21.5) 64 (49.2) 38 (29.2)	- - -	n.a. n.a. n.a.	n.a. n.a. n.a.
MELD score, *median (IQR)*	15.0 (10.0–19.0)	-	n.a.	n.a.
Mortality on waiting list, *n* (%)	29 (22.3)	-	n.a.	n.a.
History of cardiovascular disease, *n (%)*	6 (4.8)	28 (7.6)	0.282	0.067
History of diabetes, *n (%)*	36 (28.6)	106 (28.6)	0.987	0.055
Glucose lowering drugs, *n (%)*	35 (31.8)	79 (21.8)	0.031	0.697
Lipid lowering drugs, *n (%)*	19 (17.3)	87 (24.0)	0.140	0.019
**Blood tests**				
ALT (U/L), *median (IQR)*	40.0 (28.0–60.0)	25.0 (18.0–36.5)	<0.001	0.002
AST (U/L), *median (IQR)*	54.0 (44.0–84.0)	25.0 (20.0–33.0)	<0.001	<0.001
GGT (U/L), *median (IQR)*	95.5 (48.3–150.8)	39.0 (21.0–89.0)	<0.001	0.483
ALP (U/L), *mean ± SD*	140.5 (98.3–215.3)	87.0 (69.3–129.8)	<0.001	0.003
Bilirubin direct, *median (IQR)*	27.0 (16.0–85.0)	9.0 (7.0–12.0)	<0.001	0.002
Bilirubin total, *median (IQR)*	40.0 (23.0–99.0)	10.0 (7.0–14.0)	<0.001	<0.001
Albumin (g/L), *median (IQR)*	31.0 (27.0–36.0)	44.0 (42.0–46.0)	<0.001	<0.001
HbA1c (mmol/mol), *median (IQR)*	32.0 (26.0–39.5)	36.0 (32.0–43.0)	0.001	0.001
HbA1c (%), *median (IQR)*	5.1 (4.5–5.8)	5.4 (5.1–6.1)	0.001	0.001
Fasting glucose (mmol/L), *median (IQR)*	6.4 (5.0–8.5)	5.7 (5.2–6.8)	0.133	0.712
**Lipids and lipoproteins**				
Total cholesterol (mmol/L), *median (IQR)*	3.2 (2.5–4.1)	4.2 (3.6–4.9)	<0.001	<0.001
Non-HDL cholesterol (mmol/L), *median (IQR)*	2.2 (1.7–3.0)	2.8 (2.2–3.4)	<0.001	0.003
HDL cholesterol (mmol/L), *median (IQR)*	0.9 (0.4–1.2)	1.3 (1.1–1.7)	<0.001	<0.001
LDL cholesterol (mmol/L), *median (IQR)*	1.8 (1.2–2.3)	2.1 (1.7–2.5)	<0.001	0.031
Triglycerides (mmol/L), *median (IQR)*	0.7 (0.5–1.1)	1.3 (1.0–1.8)	<0.001	<0.001
ApoB (g/L), *median (IQR)*	63.0 (48.0–83.8)	74.0 (59.0–89.0)	<0.001	0.182
ApoA-I (g/L), *median (IQR)*	57.5 (31.8–82.3)	128.0 (111.0–146.0)	<0.001	<0.001
TRLP (nmol/L), *median (IQR)*	94.3 (39.6–147.0)	127.7 (85.4–182.6)	<0.001	0.001
Very large TRLP (nmol/L), *median (IQR)*	0.0 (0.0–0.0)	0.0 (0.0–0.1)	<0.001	0.173
Large TRLP (nmol/L), *median (IQR)*	0.3 (0.0–1.4)	2.0 (0.4–5.8)	<0.001	<0.001
Medium TRLP (nmol/L), *median (IQR)*	2.1 (0.0–8.0)	15.4 (7.0–26.3)	<0.001	<0.001
Small TRLP (nmol/L), *median (IQR)*	17.3 (1.6–34.0)	33.4 (17.0–57.5)	<0.001	<0.001
Very small TRLP (nmol/L), *median (IQR)*	62.3 (18.2–117.2)	65.5 (35.7–107.4)	0.370	0.836
TRL size (nm), *median (IQR)*	41.5 (35.4–49.4)	45.9 (41.6–51.8)	<0.001	<0.001
LDLP (nmol/L), *median (IQR)*	1026.5 (745.3–1477.3)	1234.5 (1013.5–1494.8)	<0.001	0.316
Large LDLP (nmol/L), *median (IQR)*	438.0 (208.8.3–671.8)	292.5 (139.8–496.0)	<0.001	<0.001
Medium LDLP (nmol/L), *median (IQR)*	0.0 (0.0–155.3)	139.5 (0.0–396.0)	<0.001	<0.001
Small LDLP (nmol/L), *median (IQR)*	256.0 (98.8–398.5)	637.0 (403.3–860.5)	<0.001	<0.001
LDL size (nm), *median (IQR)*	21.7 (21.1–22.0)	21.1 (20.7–21.4)	<0.001	<0.001
Total HDLP (µmol/L), *median (IQR)*	6.5 (4.0–10.0)	19.5 (17.1–21.6)	<0.001	<0.001
Large HDLP (µmol/L), *median (IQR)*	2.3 (1.0–3.8)	2.1 (1.2–3.6)	0.415	0.930
Medium HDLP (µmol/L), *median (IQR)*	0.5 (0.0–2.0)	4.8 (3.6–6.3)	<0.001	<0.001
Small HDLP (µmol/L), *median (IQR)*	3.0 (1.2–5.5)	12.1 (9.6–14.1)	<0.001	<0.001
HDL size (nm), *median (IQR)*	10.4 (9.3–11.3)	9.2 (8.8–9.7)	<0.001	<0.001
HDL subspecies H1P, (µmol/L), *median (IQR)* H2P, (µmol/L), *median (IQR)* H3P, (µmol/L), *median (IQR)* H4P, (µmol/L), *median (IQR)* H5P, (µmol/L), *median (IQR)* H6P, (µmol/L), *median (IQR)* H7P, (µmol/L), *median (IQR)*	0.0 (0.0–0.2) 2.8 (0.9–5.1) 0.1 (0.0–0.8) 0.2 (0.0–0.9 )0.1 (0.0–0.6) 0.4 (0.0–1.1) 0.8 (0.0–2.5)	1.9 (0.5–3.4) 9.6 (7.8–11.5) 3.1 (1.6–4.3) 1.7 (1.0–2.4) 0.5 (0.2–1.0) 0.7 (0.3–1.7) 0.4 (0.1–1.0)	<0.001 <0.001 <0.001 <0.001 <0.001 <0.001 0.044	<0.001 <0.001 <0.001 <0.001 <0.001 <0.001 <0.001
LP-Z present, *n (%)* LP-Z (nmol/L)*, median (IQR)*	40 (30.8%) 0 (0.0–666.4)	3 (0.8%) 0 (0.0–0.0)	<0.001 <0.001	<0.001 <0.001
**Primary liver diseases**				
Storage disease, *n (%)*	4 (3.1)	33 (8.9)	0.031	n.a.
Autoimmune hepatitis, *n (%)*	11 (8.5)	19 (5.1)	0.165	n.a.
Cholestatic liver disease (PSC/PBC), *n (%)*	34 (26.2)	99 (26.6)	0.919	n.a.
Viral, *n (%)*	12 (9.2)	39 (10.5)	0.684	n.a.
Alcohol, *n (%)*	29 (22.3)	49 (13.2)	0.013	n.a.
MAFLD, *n (%)*	33 (25.4)	34 (9.1)	<0.001	n.a.
Vascular, *n (%)*	2 (1.5)	1 (0.3)	0.166	n.a.
Malignancy, *n (%)*	0 (0)	5 (1.3)	0.334	n.a.
Other, *n (%)*	5 (3.8)	93 (25.0)	<0.001	n.a.

* *p*-value adjusted for age, sex, and primary liver disease. Fasting glucose was available in 299 patients. Data are given in number with percentages (%) or median with interquartile ranges (IQR). Abbreviations: ALP, alkaline phosphatase; ALT, aminotransferase; ApoA, apolipoprotein A; ApoB, apolipoprotein B; AST, aspartate aminotransferase; BMI, body mass index; GGT, gamma-glutamyltransferase; HbA1c, glycated hemoglobin; HDL, high density lipoproteins; HDLP, high density lipoprotein particles; IQR, interquartile range; LDL, low density lipoproteins; LDLP, low density lipoprotein particles; MAFLD, metabolic associated fatty liver disease; MELD score, Model for End-Stage Liver Disease; MELD-Na score, Model for End-Stage Liver Disease natrium; PBC, primary biliary cholangitis; PSC, primary sclerosing cholangitis; TRL, triglyceride-rich lipoproteins; TRLP, triglyceride-rich lipoprotein particles.

**Table 2 jcm-11-01223-t002:** Lipid profile of pre-transplant cirrhotic patients and orthotopic liver transplant (OLT) recipients compared with the PREVEND population.

	Comparison of Pre-Transplant Cirrhotic Patients (N = 130) with the PREVEND Population (N = 6027)	Comparison of OLT Recipients (N = 372) with the PREVEND Population (N = 6027)
**Lipids and lipoproteins**		
Total cholesterol (mmol/L), *median (IQR)*	−1.827 (−1.989 to −1.665) *	−1.000 (−1.097 to −0.904) *
Non-HDL cholesterol (mmol/L), *median (IQR)*	−1.514 (−1.676 to −1.352) *	−1.157 (−1.253 to −1.060) *
HDL cholesterol (mmol/L), *median (IQR)*	−1.096 (−1.249 to −0.944) *	0.508 (0.415 to 0.600) *
LDL cholesterol (mmol/L), *median (IQR)*	−1.684 (−1.844 to −1.525) *	−1.426 (−1.521 to −1.331) *
Triglycerides (mmol/L), *median (IQR)*	−0.552 (−0.724 to −0.379) *	0.141 (0.038 to 0.245) *
ApoB (mg/dL), *median (IQR)*	−0.803 (−0.971 to −0.635) *	−0.600 (−0.699 to −0.501) *
ApoA-I (mg/dL), *median (IQR)*	−2.815 (−2.961 to −2.669) *	0.001 (−0.086 to 0.089)
TRLP (nmol/L), *median (IQR)*	−0.644 (−0.813 to −0.476) *	−0.235 (−0.335 to −0.134) *
Very large TRLP (nmol/L), *median (IQR)*	−0.088 (−0.263 to 0.087)	0.025 (−0.079 to 0.130)
Large TRLP (nmol/L), *median (IQR)*	−0.609 (−0.779 to −0.439) *	−0.023 (−0.127 to 0.081)
Medium TRLP (nmol/L), *median (IQR)*	−0.732 (−0.901 to −0.564) *	0.149 (0.047 to 0.251) *
Small TRLP (nmol/L), *median (IQR)*	−0.740 (−0.913 to −0.567) *	−0.353 (−0.457 to −0.249) *
Very small TRLP (nmol/L), *median (IQR)*	0.056 (−0.117 to 0.228)	−0.046 (−0.149 to 0.056)
TRL size (nm), *median (IQR)*	−0.423 (−0.596 to −0.250) *	0.116 (0.013 to 0.219) **
LDLP (nmol/L), *median (IQR)*	−0.781 (−0.949 to −0.612) *	−0.606 (−0.705 to −0.507) *
Large LDLP (nmol/L), *median (IQR)*	0.628 (0.463 to 0.792) *	−0.089 (−0.186 to 0.009)
Medium LDLP (nmol/L), *median (IQR)*	−0.950 (−1.121 to −0.778) *	−0.531 (−0.634 to −0.428) *
Small LDLP (nmol/L), *median (IQR)*	−0.833 (−1.006 to −0.661) *	−0.014 (−0.118 to 0.090)
LDL size (nm), *median (IQR)*	0.833 (0.667 to 1.000) *	−0.051 (−0.150 to 0.048)
Total HDLP (µmol/L), *median (IQR)*	−3.990 (−4.129 to −3.851) *	−0.473 (−0.556 to −0.389) *
Large HDLP (µmol/L), *median (IQR)*	0.518 (0.362 to 0.674) *	0.631 (0.536 to 0.727) *
Medium HDLP (µmol/L), *median (IQR)*	−1.619 (−1.777 to −1.460) *	−0.030 (−0.126 to 0.065)
Small HDLP (µmol/L), *median (IQR)*	−3.289 (−3.435 to −3.143) *	−0.731 (−0.820 to −0.642) *
HDL size (nm), *median (IQR)*	0.371 (2.436 to 2.733) *	0.655 (0.571 to 0.738) *
HDL subspecies		
H1P, (µmol/L), *median (IQR)* H2P, (µmol/L), *median (IQR)* H3P, (µmol/L), *median (IQR)* H4P, (µmol/L), *median (IQR)* H5P, (µmol/L), *median (IQR)* H6P, (µmol/L), *median (IQR)* H7P, (µmol/L), *median (IQR)*	−1.719 (−1.881 to −1.557) * −2.839 (−2.995 to −2.682) * −1.353 (−1.517 to −1.189) * −1.026 (−1.190 to −0.862) * −0.098 (−0.264 to 0.069) −0.217 (−0.379 to −0.055) * 1.686 (1.536 to 1.836) *	−0.681 (−0.780 to −0.582) * −0.424 (−0.519 to −0.330) * −0.102 (−0.202 to −0.002) ** 0.108 (0.008 to 0.208) ** 0.596 (0.492 to 0.699) * 0.239 (0.140 to 0.338) * 0.610 (0.520 to 0.700) *
LP-Z (nmol/L)*, median (IQR)*	3.176 (3.017 to 3.336) *	0.031 (0.008 to 0.055) *

* *p* < 0.01 or less. ** *p* < 0.05. Data represent regression coefficients of the standard deviation scores (*Z*-scores) adjusted for age and sex with corresponding 95% confidence intervals. Comparison of the pre-transplant cirrhotic, OLT recipients with the control group PREVEND. Abbreviations: ApoA, apolipoprotein A; ApoB, apolipoprotein B; HDL, high density lipoproteins; HDLP, high density lipoprotein particles; IQR, interquartile range; LDL, low density lipoproteins; LDLP, low density lipoprotein particles; LP-Z, lipoprotein Z; TRL, triglyceride-rich lipoproteins; TRLP, triglyceride-rich lipoprotein particles.

**Table 3 jcm-11-01223-t003:** Clinical and biochemical characteristics of LP-Z in 130 pre-transplant cirrhotic patients from TransplantLines.

	LP-Z Not Detectable N = 90 (69.2%)	LP-Z Detectable N = 40 (30.8%)	*p*-Value	Adjusted *p*-Value *
**Baseline characteristics**				
Sex: men/women, *n (%)*	60 (66.7)/30 (33.3)	25 (62.5)/15 (37.5)	0.645	n.a.
Age (years), *median (IQR)*	60.0 (54.0–65.3)	58.5 (51.0–64.0)	0.296	n.a.
BMI (kg/m^2^), *median (IQR)*	28.2 (24.6–31.1)	26.4 (23.2–31.0)	0.239	0.932
BMI				
Normal; ≤25 kg/m^2^, *n (%)* Overweight; 25–30 kg/m^2^, *n (%)* Obese; ≥30 kg/m^2^, *n (%)*	17 (26.2) 25 (38.5) 23 (35.4)	11 (42.3) 7 (26.9) 8 (30.8)	0.131 0.298 0.675	0.494 0.132 0.255
Smoking, *n (%)*	13 (20.0)	3 (15.0)	0.617	0.418
Child Pugh Turcotte classification Child Pugh Turcotte A, *n (%)* Child Pugh Turcotte B, *n (%)* Child Pugh Turcotte C, *n (%)*	24 (26.7) 53 (58.9) 13 (14.4)	4 (10.0) 11 (27.5) 25 (62.5)	0.038 0.001 <0.001	0.035 0.001 <0.001
MELD score, *median (IQR)*	13.0 (9.8–16.0)	19.5 (17.0–22.0)	<0.001	<0.001
Mortality on waiting list, *n* (%)	18 (20.0)	11 (27.5)	0.150	0.999
History of cardiovascular disease, *n (%)*	6 (6.9)	0 (0.0)	0.176	0.999
History of diabetes, *n (%)*	28 (32.2)	8 (20.5)	0.180	0.252
Glucose lowering drugs, *n (%)*	28 (37.8)	7 (19.4)	0.052	0.069
Lipid lowering drugs, *n (%)*	13 (17.6)	6 (16.7)	0.907	0.775
**Blood tests**				
ALT (U/L), *median (IQR)*	36.0 (28.0–49.5)	53.5 (33.5–112.3)	0.006	0.031
AST (U/L), *median (IQR)*	50.0 (37.0–62.3)	88.0 (54.0–152.8)	<0.001	0.069
GGT (U/L), *median (IQR)*	97.5 (52.5–152.0)	69.5 (40.5–141.5)	0.378	0.630
ALP (U/L), *mean ± SD*	130.5 (92.0–167.5)	218.5 (104.3–298.0)	0.005	0.004
Bilirubin direct, *median (IQR)*	18.0 (13.0–29.0)	162.0 (51.0–228.0)	<0.001	0.002
Bilirubin total, *median (IQR)*	28.0 (18.0–52.5)	162.5 (75.5–257.3)	<0.001	<0.001
Albumin (g/L), *median (IQR)*	32.0 (27.3–36.0)	28.0 (25.3–36.0)	0.111	0.169
HbA1c (mmol/mol), *median (IQR)*	34.0 (27.0–46.0)	27.0 (18.0–33.0)	0.018	0.095
HbA1c (%), *median (IQR)*	5.3 (4.7–6.4)	4.6 (3.8–5.2)	0.017	0.093
Fasting glucose (mmol/L), *median (IQR)*	6.4 (5.0–8.0)	6.6 (4.9–9.8)	0.554	0.326
**Lipids and lipoproteins**				
Total cholesterol (mmol/L), *median (IQR)*	3.4 (2.8–4.1)	2.4 (1.6–4.2)	0.001	0.044
Non-HDL cholesterol (mmol/L), *median (IQR)*	2.3 (1.9–3.0)	2.1 (1.4–3.9)	0.504	0.356
HDL cholesterol (mmol/L), *median (IQR)*	1.1 (0.8–1.3)	0.2 (0.1–0.5)	<0.001	<0.001
LDL cholesterol (mmol/L), *median (IQR)*	1.9 (1.4–2.2)	1.7 (1.1–2.3)	0.409	0.915
Triglycerides (mmol/L), *median (IQR)*	0.6 (0.5–1.0)	0.9 (0.5–1.3)	0.101	0.152
ApoB (mg/dL), *median (IQR)*	57.0 (46.8–73.5)	94.0 (50.5–123.0)	<0.001	<0.001
ApoA-I (mg/dL), *median (IQR)*	68.5 (53.0–88.0)	20.5 (11.3–34.0)	<0.001	<0.001
TRLP (nmol/L), *median (IQR)*	112.0 (69.6–147.0)	28.4 (4.7–163.6)	0.001	0.220
Very large TRLP (nmol/L), *median (IQR)*	0.0 (0.0–0.1)	0.0 (0.0–0.0)	0.010	0.310
Large TRLP (nmol/L), *median (IQR)*	0.3 (0.0–1.6)	0.1 (0.0–1.3)	0.208	0.358
Medium TRLP (nmol/L), *median (IQR)*	2.0 (0.0–6.8)	2.7 (0.0–13.3)	0.215	0.138
Small TRLP (nmol/L), *median (IQR)*	20.3 (11.2–33.8)	0.6 (0.0–29.3)	0.001	0.859
Very small TRLP (nmol/L), *median (IQR)*	74.6 (38.8–118.4)	4.6 (0.0–110.6)	<0.001	0.100
TRL size (nm), *median (IQR)*	40.9 (34.7–48.4)	45.6 (36.2–55.7)	0.154	0.756
LDLP (nmol/L), *median (IQR)*	956.0 (721.5–1166.3)	1715.0 (968.5–2256.3)	<0.001	<0.001
Large LDLP (nmol/L), *median (IQR)*	472.5 (337.8–671.8)	235.5 (20.5–727.3)	0.007	0.274
Medium LDLP (nmol/L), *median (IQR)*	16.5 (0.0–234.5)	0.0 (0.0–0.0)	<0.001	0.004
Small LDLP (nmol/L), *median (IQR)*	277.0 (113.0–425.0)	182.0 (26.8–394.0)	0.108	0.298
LDL size (nm), *median (IQR)*	21.7 (21.3–22.0)	21.3 (20.0–22.1)	0.033	<0.001
Total HDLP (µmol/L), *median (IQR)*	8.1 (5.8–11.5)	3.3 (1.6–4.8)	<0.001	<0.001
Large HDLP (µmol/L), *median (IQR)*	3.1 (1.9–3.9)	0.7 (0.0–1.5)	<0.001	<0.001
Medium HDLP (µmol/L), *median (IQR)*	0.9 (0.2–3.0)	0.0 (0.0–0.3)	<0.001	<0.001
Small HDLP (µmol/L), *median (IQR)*	3.4 (1.8–6.1)	2.1 (0.4–3.2)	0.001	0.001
HDL size (nm), *median (IQR)*	10.8 (9.7–11.5)	9.5 (8.1–10.3)	<0.001	<0.001
HDL subspecies H1P, (µmol/L), *median (IQR)* H2P, (µmol/L), *median (IQR)* H3P, (µmol/L), *median (IQR)* H4P, (µmol/L), *median (IQR)* H5P, (µmol/L), *median (IQR)* H6P, (µmol/L), *median (IQR)* H7P, (µmol/L), *median (IQR)*	0.0 (0.0–0.4) 3.3 (1.3–5.6) 0.4 (0.0–1.2) 0.5 (0.0–1.3) 0.1 (0.0–0.6) 0.6 (0.0–1.3) 1.8 (0.3–2.9)	0.0 (0.0–0.1) 1.7 (0.0–2.8) 0.0 (0.0–0.1) 0.0 (0.0–0.1) 0.0 (0.0–0.6) 0.0 (0.0–0.5) 0.0 (0.0–0.3)	0.264 <0.001 <0.001 <0.001 0.699 <0.001 <0.001	0.962 <0.001 0.004 0.001 0.377 <0.001 <0.001
LP-Z (nmol/L)*, median (IQR)*	-	1354.5 (725.7–2080.9)	n.a.	n.a.
**Primary liver diseases**				
Storage disease, *n (%)*	4 (4.4)	0 (0)	0.311	n.a.
Autoimmune hepatitis, *n (%)*	8 (8.9)	3 (7.5)	1.00	n.a.
Cholestatic liver disease (PSC/PBC), *n (%)*	18 (20.0)	16 (40.0)	0.017	n.a.
Viral, *n (%)*	9 (10.0)	3 (7.5)	0.754	n.a.
Alcohol, *n (%)*	22 (24.4)	7 (17.5)	0.380	n.a.
MAFLD, *n (%)*	24 (26.7)	9 (22.5)	0.614	n.a.
Vascular, *n (%)*	2 (2.2)	0 (0.0)	1.00	n.a.
Malignancy, *n (%)*	0 (0)	0 (0)	n.a.	n.a.
Other, *n (%)*	3 (3.3)	2 (5.0)	0.643	n.a.

* *p*-value adjusted for age, sex, and primary liver disease. Data are given in number with percentages (%) or median with interquartile ranges (IQR). Abbreviations: ALP, alkaline phosphatase; ALT, aminotransferase; ApoA, apolipoprotein A; ApoB, apolipoprotein B; AST, aspartate aminotransferase; BMI, body mass index; GGT, gamma-glutamyltransferase; HbA1c, glycated hemoglobin; HDL, high density lipoproteins; HDLP, high density lipoprotein particles; IQR, interquartile range; LDL, low density lipoproteins; LDLP, low density lipoprotein particles; LP-Z, lipoprotein Z; MAFLD, metabolic associated fatty liver disease; MELD, Model for End-Stage Liver Disease; MELD-Na, Model for End-Stage Liver Disease natrium; PBC, primary biliary cholangitis; PSC, primary sclerosing cholangitis; TRL, triglyceride-rich lipoprotein; TRLP, triglyceride-rich lipoprotein particles.

## Data Availability

The data presented in this study are available on request from the corresponding author.

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
