# Peer review of "Profoundly Disturbed Lipoproteins in Cirrhotic Patients: Role of Lipoprotein-Z, a Hepatotoxic LDL-like Lipoprotein"

_jcm, 2022, doi:10.3390/jcm11051223_

Round 1
Reviewer 1 Report
The study carried out by van den Berg et al. clearly suggests that LP-Z is a novel independent biomarker of worse survival in cirrhotic patients. The methods used in this study are excellent and reliable, and the findings are striking and medically very valuable. The manuscript is well-written and easy to read. I have only minor comments.
1. Considering the indication for OLT, Child-Pugh and MELD scores might be associated with the etiology of primary liver disease. Therefore, it would be better to show the relationships between LP-Z concentrations and these scores taking into account the primary liver disease, in addition to Figure 2.
2. Lines 113-114.
Please explain why one of 15 patients was not included in the OLT recipient group.
3. Line 437
Please correct the error: it was it was suggested
Author Response
The study carried out by van den Berg et al. clearly suggests that LP-Z is a novel independent biomarker of worse survival in cirrhotic patients. The methods used in this study are excellent and reliable, and the findings are striking and medically very valuable. The manuscript is well-written and easy to read. I have only minor comments.
Response: We very much appreciate the careful reading of our manuscript by the reviewer and the valuable remarks which have resulted in improvements in our manuscript.
Minor comments of the reviewer
- Considering the indication for OLT, Child-Pugh and MELD scores might be associated with the aetiology of primary liver disease. Therefore, it would be better to show the relationships between LP-Z concentrations and these scores taking into account the primary liver disease, in addition to Figure 2.
Response: This point of the reviewer is well taken. We have now also carried out multivariable regression analyses which did not only include age, sex, CPT classification and MELD scores but also the aetiologies of primary liver disease as cause of cirrhosis. This analysis is presented as Supplementary Table 3 of the revised manuscript and shows that measurable LP-Z is independently and positively associated with MELD scores as well as with cholestatic liver disease and malignancy as underlying aetiology.
- Lines 113-114.
Please explain why one of 15 patients was not included in the OLT recipient group.
Response: One patient was not included in the post OLT group because this patient developed cirrhosis post OLT.
- Line 437
Please correct the error: it was it was suggested
Response: We thank the reviewer of pointing to the error. In the revised paper we have corrected this duplicate wording.
Reviewer 2 Report
Interesting and novel theme
Author Response
Response: We appreciate the careful reading of our manuscript by the reviewer. As general feedback the reviewer states that the introduction might benefit of some improvements. In the revised version of our manuscript we have made several changes in the introduction section to improve its readability.
The reviewer states that our manuscript deals with an interesting and novel theme.
Response: We much appreciate the positive judgment of our paper by the reviewer. No further comments are given by the reviewer that need to be answered.